# An application of node and edge nonlinear hypergraph centrality to a protein complex hypernetwork

**Sarah Lawson** *, Diane Donovan, James Lefevre

ARC Centre of Excellence, Plant Success in Nature and Agriculture, School of Mathematics and Physics, The University of Queensland, Brisbane, Queensland, Australia

* sarah.lawson@uq.edu.au

**Data Availability Statement:** All code and data used in this work is publicly available via the online repository https://github.com/LawsonSarah/PCH_node_edge_centrality.

## Abstract

The use of graph centrality measures applied to biological networks, such as protein interaction networks, underpins much research into identifying key players within biological processes. This approach however is restricted to dyadic interactions and it is well-known that in many instances interactions are polyadic. In this study we illustrate the merit of using hypergraph centrality applied to a hypernetwork as an alternative. Specifically, we review and propose an extension to a recently introduced node and edge nonlinear hypergraph centrality model which provides mutually dependent node and edge centralities. A *Saccharomyces Cerevisiae* protein complex hypernetwork is used as an example application with nodes representing proteins and hyperedges representing protein complexes. The resulting rankings of the nodes and edges are considered to see if they provide insight into the essentiality of the proteins and complexes. We find that certain variations of the model predict essentiality more accurately and that the degree-based variation illustrates that the centrality-lethality rule extends to a hypergraph setting. In particular, through exploitation of the models flexibility, we identify small sets of proteins densely populated with essential proteins. One of the key advantages of applying this model to a protein complex hypernetwork is that it also provides a classification method for protein complexes, unlike previous approaches which are only concerned with classifying proteins.

## Introduction

The origins of graph theory date back to the mid 18th century [1] with the field advancing over the next three centuries to become an area of significant interest drawing on and influencing many other areas of mathematics. One of the many notable advances was the introduction of hypergraphs, formally introduced by Berge in 1970 [2]. The (hyper)edges of a hypergraph can contain an arbitrary number of vertices, as opposed to a graph whose edges consist of at most two vertices, thus a hypergraph can model more complex systems. The analysis of large and complex datasets represented by hypergraphs, also known as hypernetworks in the applied setting, has led to numerous intriguing results in recent years. Examples include the analysis of

**Funding:** This work was supported by The Australian Research Council, through the Centre of Excellence for Plant Success in Nature and Agriculture (CE200100015) https://www.plantsuccess.org/. Author DD is CI in the CoE. The CoE and ARC had no direct role in study design, data collection and analysis, decision to publish, or preparation of the manuscript.

**Competing interests:** The authors have declared that no competing interests exist.

polyadic interactions between proteins in humans [3] and multi-modal biomarkers for Alzheimer's disease [4].

Following the introduction of hypergraphs, researchers started to look at extending graph theoretical concepts to provide insights into this new area. One such area of interest is centrality measures which rank vertices and edges by some defined 'importance' or 'centrality'. This has been extensively researched in the context of graphs with each measure having their own merit based on what constitutes as 'important' in the given problem. For example, degree centrality ranks vertices by their degree in descending order. Of particular interest is eigenvector centrality which "accords each vertex a centrality that depends on both the number and quality of its connections" [5]. In 1972, Bonacich [6] introduced graph eigenvector centrality utilising the eigenvectors of the adjacency matrix. When ordered by descending size, the entries in the eigenvector give a ranking of the graph vertices. However, although Bonacich is held to have introduced the measure formally, the concept had appeared earlier in different disciplines. Edmund Landaus's ranking of chess players in 1895 [7] is thought to be the first documented usage of an eigenvector centrality type measure.

Eigenvector centrality can also be applied to the 'importance' of edges and further extended to the 'importance' of edges and vertices being mutually dependent on each other. This latter approach was introduced by Bonacich through "simultaneous eigenvector centrality scores" [8]. Faust considered these centralities in the context of affiliation networks, a social network which consists of *actors* and subsets of actors called *events* [9]. These centrality scores were the foundation of linear eigenvector centrality for uniform hypergraphs which was formally defined in 2004 [10]. Aiming to overcome the limitations of a linear centrality, Benson [11] proposed a non-linear eigenvector centrality for uniform hypergraphs utilising adjacency tensors. In 2021, Tudisco and Higham [12] took this further and introduced a model which not only combined the aforementioned centralities but extended them to nonlinear centralities for nonuniform hypergraphs. It is this model that we focus on in this research. The authors refer to their model as the *Nonlinear Singular Vector Centrality Model* in their paper and we adopt this terminology using the abbreviation NSVC. As detailed in the Methods section, the model incorporates a set of four functions, $f$, $g$, $\varphi$, $\psi$. The nature of the mutually dependent relationship between the nodes and the edges is dictated by these user-defined functions. For example, we can define $f$, $g$, $\varphi$, $\psi$ in such a way that the ranking of a node is based solely on the highest ranked edge to which it belongs and, simultaneously, an edges rank is based solely on the highest ranking node it contains. The model outputs two vectors, $x$ and $y$, whose entries capture the individual node and edge centrality values respectively. A larger centrality value indicates a more important node or edge thus the ordered vectors provide a ranking of the nodes and edges. (Both the terms *vertex* and *node* are in common use. The former more often in the context of theoretical hypergraphs and the latter in applied hypernetworks. They are used interchangeably in this paper).

When defining an eigenvector type centrality measure, one of the key considerations is the conditions under which a unique positive eigenvector exists for the defined problem. Classical Perron Frobenius Theorem (PFT) concerns itself with proving the existence of unique, positive eigenvectors for a positive matrix as proposed by Perron in 1907 [13] and extended by Frobenius in 1912 to positive eigenvectors of a nonnegative irreducible matrix [14]. In recent years, PFT has been extended to many areas. Of particular interest is its extension to tensors [15] and multihomogeneous mappings [16], the former being used to evidence the uniqueness and existence of certain tensor eigenvector centralities for uniform hypergraphs [11] and the latter used by Tudisco and Higham [12] to prove their existence and uniqueness theorem under certain conditions. Reference to the existence and particularly the uniqueness of a

solution is surprisingly often omitted when eigenvector centrality is applied to real-world scenarios.

One of the primary objectives of this work is to showcase the adaptability of the NSVC model and it's potential as an analytic tool to identify important nodes and edges in hypergraphs representing biological networks. We apply the model to a hypernetwork representing protein complexes in *Saccharomyces Cerevisiae*, commonly known as brewers yeast, and demonstrate that the resulting rankings can be used to predict essential proteins and complexes. Essential proteins carry out essential cellular functions and the deletion of such proteins leads to cell inviability. A protein complex is a group of proteins interacting with each other to enact cellular functions. Definitions of an essential protein complex vary but are often based on the number or proportion of essential proteins within the complex. This is discussed further within the NSVC application to S.Cerevisiae section. Protein interactions are often represented as a dyadic Protein Interaction Network (PIN) where the nodes represent proteins and the edges represent the interaction between two proteins. In a 2001 article, Jeong et al [17] stated "the most highly connected proteins in the cell are the most important for its survival". This is known as the centrality-lethality rule, namely that nodes with higher degree centrality are more likely to be essential. This short paper has been cited over 6000 times and has motivated many researchers in the following decades to utilise centrality as a tool to identify essential proteins in PINs. One of the earlier papers building on this idea was by Estrada in 2006 [18] who applied the degree, closeness, betweenness, eigenvector, information and subgraph centralities to a yeast PIN and showed that each performed better than random selection in identifying essential proteins. Extensive progress improving on these preliminary results has been made over the years not only incorporating network topology but also including biological information and machine learning techniques to classify essential proteins. Numerous comprehensive reviews are available detailing these approaches, such as [19–22]. Many of these approaches have incorporated protein complex information into their centrality measures, such as ECC [23], PCSD [24], LBCC [25], CENC [26], LIDC [27] and UC [28]. In this work we propose the use of hypergraph centrality measures, specifically the NSVC model, applied to a protein complex hypernetwork (PCH) with nodes representing proteins and edges representing complexes [3, 29, 30] as an alternative to PIN-based approaches. Using a PCH retains information that would otherwise be lost in a PIN. Significantly, our proposal also provides a novel ranking mechanism for protein complexes.

In this paper we first consider what it means for a node or edge to rank higher in each of Tudisco and Higham's proposed function sets for their model and then we propose a further generalisation. We then apply this generalised model to the S.Cervisiae PCH as an example of its use. We consider whether the resulting rankings place nodes and edges representing essential proteins and complexes higher than those representing nonessential proteins and complexes. Ultimately, we show that the NSVC model can be customised to improve its predictive capabilities and provide an alternative platform for future researchers to build on.

## Methods

### Hypergraph definitions

A *hypergraph* is a tuple $H = (V, E)$ where $V$ is a set of vertices and $E$ is a set of *(hyper)edges*. An edge in an undirected hypergraph is defined as a set $e = \{v_1, \ldots, v_k\} \subseteq V$ for some $1 \leq k \leq |V|$. A *k-uniform hypergraph* is a hypergraph where every edge contains exactly $k$ vertices.

A *hyperpath* in a hypergraph $H$ is an ordered list of vertices and edges $(v_0, e_1, v_1, e_2, \ldots, e_m, v_m)$ such that $v_{i-1}, v_i \in e_i$ for $1 \leq i \leq m$ and $v_j \neq v_k$ for $0 \leq j < k \leq m$. Edges are not necessarily

distinct. A hypergraph $H$ is *connected* if for all $v_i, v_j \in V$ there exists a hyperpath between $v_i$ and $v_j$ in $H$.

Two vertices are *adjacent* if they belong to the same edge and two edges are *adjacent* if they have a non-empty intersection. A vertex $i$ in an edge $e$ is said to be *incident* to $e$. Similarly, an edge $e$ containing a node $i$ is said to be *incident* to $i$. An *incidence matrix* for a hypergraph $H = (V, E)$ where $|V| = n$ and $|E| = m$ is an $n \times m$ matrix $B = [b_{ie}]$ such that:

$$b_{ie} = \begin{cases} 1, & \text{if } i \in e, \\ 0, & \text{otherwise.} \end{cases}$$

Given a hypergraph $H$, the *weighted node degree $d_i^V$* of a vertex $i \in V$ is $d_i^V = \sum_{e \in E} b_{ie} w(e)$ [2] with $w : E \to \mathbb{R}_{\geq 0}$ being the edge weight function. If a hypergraph is unweighted then $d_i^V = \sum_{e \in E} b_{ie}$, the number of edges to which vertex $i$ belongs. The *weighted edge degree $d_e^E$* of an edge $e \in E$ is $d_e^E = \sum_{i \in V} b_{ie} n(i)$ with $n : V \to \mathbb{R}_{\geq 0}$ being the node weight function. If a hypergraph is unweighted then $d_e^E = \sum_{i \in V} b_{ie}$, the cardinality of the edge.

## Tudisco and Higham's NSVC model [12]

Consider a hypergraph $H$ with incidence matrix $B$, functions $f, g, \varphi, \psi : \mathbb{R}_{\geq 0} \to \mathbb{R}_{\geq 0}$ acting component wise and diagonal matrices $N$ and $W$ containing hypergraph node and edge weights respectively. The node and edge *Nonlinear Singular Vector Centralities* $\boldsymbol{x}, \boldsymbol{y} > 0$ respectively together with associated scalars $\lambda, \mu > 0$ are a solution of:

$$\begin{cases} \lambda \boldsymbol{x} = g(BWf(\boldsymbol{y})) \\ \mu \boldsymbol{y} = \psi(B^T N \varphi(\boldsymbol{x})) \end{cases} \tag{1}$$

with $\|\boldsymbol{x}\| = \|\boldsymbol{y}\| = 1$. The $i_{th}$ entry of $\boldsymbol{x}$ represents the centrality $x_i$ of node $i$ and the $e_{th}$ entry of $\boldsymbol{y}$ represents the centrality $y_e$ of edge $e$. A node and edge ranking results from this model by ordering the nodes and edges by centrality in descending order. The use of the 1-norm when normalising $\boldsymbol{x}$ and $\boldsymbol{y}$ gives the proportion of centrality for each node or edge.

Tudisco and Higham calculate these centralities through a non-linear power method. We adapt their code provided in [12] for our calculations.

**Theorem 1**. *Existence and uniqueness of solutions to the NSVC model* [12]

Let $f, g, \varphi, \psi : \mathbb{R}_{\geq 0} \to \mathbb{R}_{\geq 0}$ be order preserving and homogeneous functions of degrees $\alpha$, $\beta$, $\gamma$, $\delta$ respectively. Define the coefficient $\eta = |\alpha\beta\gamma\delta|$. If either:

P1. $\eta < 1$, or

P2. $\eta = 1$, $f, g, \varphi, \psi$ are differentiable and positive functions and the bipartite graph with adjacency matrix $\begin{bmatrix} 0 & BW \\ B^T N & 0 \end{bmatrix}$ is connected,

then there exists a unique solution $\boldsymbol{x}, \boldsymbol{y} > 0$ (up to scaling) and $\lambda, \mu > 0$ for Eq 1.

Interested readers are encouraged to refer to [12] for the background information and proof of Theorem 1.

Tudisco and Higham give three examples of function sets for their model. The following summarises these sets including a brief review of their resulting rankings. A generalised

function set is subsequently proposed which incorporates two of these three function sets and allows for further variations of the model to better fit a proposed use.

**Linear function set.**

$$f(\boldsymbol{x}) = \boldsymbol{x}, g(\boldsymbol{x}) = \boldsymbol{x}, \varphi(\boldsymbol{x}) = \boldsymbol{x}, \psi(\boldsymbol{x}) = \boldsymbol{x}.$$

The resulting centrality measures after inputting these functions into Eq 1 encompass the linear node and edge centralities based on the incidence matrix for graphs proposed by Bonacich [8] and its extension to hypergraphs detailed in Bonacich et al [10]:

$$\lambda\boldsymbol{x} = BW\boldsymbol{y} \Rightarrow \lambda x_i = \sum_{\substack{e \in E \\ i \in e}} w(e)y_e, \qquad \lambda \neq 0, \tag{2}$$

$$\mu\boldsymbol{y} = B^T N\boldsymbol{x} \Rightarrow \mu y_e = \sum_{i \in e} n(i)x_i, \qquad \mu \neq 0. \tag{3}$$

In this model variation, a node which belongs to more and/or higher ranked and/or higher weighted edges ranks higher than one in fewer and smaller ranked and weighted edges. It is easily seen that $f$, $g$, $\varphi$, $\psi$ are order preserving and homogeneous. Since $\rho = |\alpha\beta\gamma\delta| = 1$ it does not meet P1, however $f$, $g$, $\varphi$, $\psi$ are differentiable and positive and thus if the bipartite graph with adjacency matrix $\begin{bmatrix} 0 & BW \\ B^T N & 0 \end{bmatrix}$ is connected then P2 is met and it follows that there exists a unique $\boldsymbol{x}$, $\boldsymbol{y}$ (up to scaling) and unique $\lambda$, $\mu > 0$ solution of the NSVC model.

**Log-exp function set.**

$$f(\boldsymbol{x}) = \boldsymbol{x}, \quad g(\boldsymbol{x}) = \boldsymbol{x}^{\frac{1}{p+1}}, \quad \varphi(\boldsymbol{x}) = ln(\boldsymbol{x}), \quad \psi(\boldsymbol{x}) = e^{(\boldsymbol{x})} \text{ for } p \in \mathbb{Z}_{>0}.$$

For a connected $k$-uniform hypergraph, this function set encompasses tensor eigenvector centralities introduced in [11]. In particular a $Z$-tensor eigenvector node centrality for $p = 1$ and a $H$-tensor eigenvector node centrality for $p = k - 1$.

Substituting $f$, $g$, $\varphi$, $\psi$ into Eq 1 gives:

$$\lambda\boldsymbol{x} = (BW\boldsymbol{y})^{\frac{1}{p+1}} \Rightarrow \lambda x_i = (\sum_{\substack{e \in E \\ i \in e}} w(e)y_e)^{\frac{1}{p+1}}, \quad \lambda \neq 0, \tag{4}$$

$$\mu\boldsymbol{y} = e^{(B^T N ln(\boldsymbol{x}))} \Rightarrow \mu y_e = e^{\left(\sum_{i \in e} n(i)ln(x_i)\right)} = \prod_{i \in e} x_i^{n(i)}, \quad \mu \neq 0. \tag{5}$$

As with the linear set, Eq 4 indicates that the more and/or higher ranked and/or weighted edges a node belongs to, the higher the node will rank. However, Eq 5 indicates that an edge will rank higher if it contains *fewer* and/or higher ranked and/or *lower* weighted nodes due to the effect of $\|\boldsymbol{x}\| = 1$.

These functions are an interesting choice as the range of $\varphi$ for the domain $\mathbb{R}_{\geq 0}$ does not fall within the specified co-domain $\mathbb{R}_{\geq 0}$. Further, neither $\varphi$ or $\psi$ are homogeneous thus the existence and uniqueness of a solution to the NSVC model with these functions cannot be guaranteed using Theorem 1. However, it is known that for connected $k$-uniform hypergraphs, positive H and Z tensor eigenvectors exist and uniqueness is guaranteed for a H eigenvector [15, 31]. Further research into alternative existence and uniqueness of solutions for functions

not meeting the required conditions would be of interest, particularly in the context of non-uniform hypergraphs.

**Max function set.**

$$f(\boldsymbol{x}) = (\boldsymbol{x}), \quad g(\boldsymbol{x}) = \boldsymbol{x}, \quad \varphi(\boldsymbol{x}) = \boldsymbol{x}^{\omega}, \quad \psi(\boldsymbol{x}) = \boldsymbol{x}^{\frac{1}{\omega}} \text{ for some } \omega \in \mathbb{Z}_{>0}.$$

The node centrality equations are given in Eq 2. Substituting $\varphi$ and $\psi$ into Eq 1 gives edge centralities:

$$\mu\boldsymbol{y} = (B^T N \boldsymbol{x}^{\omega})^{\frac{1}{\omega}} \Rightarrow \mu y_e = \left( \sum_{i \in e} \left( n(i)^{\frac{1}{\omega}} x_i \right)^{\omega} \right)^{\frac{1}{\omega}}, \quad \mu \neq 0, \tag{6}$$

which for an unweighted hypergraph and for large enough $\omega$ gives

$$\mu y_e \approx max\{x_{i_1}, \dots, x_{i_k}\}. \tag{7}$$

Again, the node centrality is linear; the more and/or higher ranked edges a node belongs to, the higher the node will rank. But for large enough $\omega$, an edges rank is only dependent on the maximum of the centralities of its incident nodes.

The same conditions as the linear case apply for existence of a unique $\boldsymbol{x}, \boldsymbol{y}$ (up to scaling) and unique $\lambda, \mu > 0$ solution of the NSVC model.

**Generalised power function set.** We now propose the following generalisation:

$$f(\boldsymbol{x}) = \boldsymbol{x}^a, g(\boldsymbol{x}) = \boldsymbol{x}^b, \varphi(\boldsymbol{x}) = \boldsymbol{x}^c, \psi(\boldsymbol{x}) = \boldsymbol{x}^d \text{ for } a, c \in \mathbb{R}_{\geq 0} \text{ and } b, d \in \mathbb{R}_{>0}.$$

For ease, we use the notation $P : (a, b, c, d)$ to represent a set of functions of this form. The values $b, d = 0$ result in the node and edge centralitites respectively being equal thus are omitted.

Substituting these functions into the NSVC model we get:

$$\lambda\boldsymbol{x} = (BW\boldsymbol{y}^a)^b \Rightarrow \lambda x_i = (\sum_{\substack{e \in E \\ i \in e}} w(e) y_e^a)^b, \quad \lambda \neq 0, \tag{8}$$

$$\mu\boldsymbol{y} = (B^T N \boldsymbol{x}^c)^d \Rightarrow \mu y_e = (\sum_{i \in e} n(i) x_i^c)^d, \quad \mu \neq 0. \tag{9}$$

from which follows

$$\mu^a \lambda x_i = (\sum_{\substack{e \in E \\ i \in e}} w(e)(\sum_{j \in e} n(j) x_j^c)^{da})^b, \tag{10}$$

$$\mu \lambda^c y_e = (\sum_{i \in e} n(i)(\sum_{\substack{f \in E \\ i \in f}} w(f) y_f^a)^{bc})^d. \tag{11}$$

The flexibility to choose variables $a, b, c, d$ allows the model to be customised to suit the application. By Theorem 1 there exists a unique solution to functions of this form if $|abcd| < 1$, or if $|abcd| = 1$ and the conditions of P2 are met.

It is clear that this function set incorporates the linear and max functions defined earlier with $P : (1, 1, 1, 1)$ and $P : (1, 1, \omega, 1/\omega)$ respectively. Significantly, it also encompasses the centralities based on the node and edge degrees with $P : (0, 1, 0, 1)$. This is of key interest in the

application of the model to the S.Cervisiae dataset due to degree centrality motivating the investigation of centrality measures as a tool to predict protein essentiality for PINs [18].

Although there are infinitely many possible sets of $a, b, c, d$ they can be combined into families when considering using the centralities solely as a ranking mechanism.

For example, the node and edge degrees are simply raised by $b$ and $d$ respectively when utilising function sets of the form $P : (0, b, 0, d)$ thus permitting the same ranking as $P : (0, 1, 0, 1)$.

Further, if we are to just consider nodes then any variation of the form $P : (0, b, c, d)$ results in the same node rankings. Similarly, any function sets of the form $P : (a, b, 0, d)$ result in the same edge rankings.

An interesting combination is $P : (\omega, 1/\omega, \omega, 1/\omega)$ which gives:

$$\lambda \boldsymbol{x} = (BW\boldsymbol{y}^\omega)^{1/\omega} \Rightarrow \lambda x_i = \Big(\sum_{\substack{e \in E \\ i \in e}} w(e) y_e^\omega\Big)^{1/\omega}, \qquad \lambda \neq 0, \tag{12}$$

$$\mu \boldsymbol{y} = (B^T N \boldsymbol{x}^\omega))^{1/\omega} \Rightarrow \mu y_e = \Big(\sum_{i \in e} n(i) x_i^\omega\Big)^{1/\omega}, \qquad \mu \neq 0. \tag{13}$$

For large enough $\omega$ and an unweighted hypergraph we get

$$\mu y_e \approx max\{x_{i_1}, \ldots, x_{i_k}\} \quad \text{where } e = \{i_1, \ldots i_k\}, \tag{14}$$

$$\lambda x_i \approx max\{y_{e_1}, \ldots, y_{e_l}\} \quad \text{where } i \in \{e_1, \ldots, e_l\}. \tag{15}$$

In our application we consider $\omega = 95$ as this value leads to a close approximation for the maximum values whilst still maintaining efficiency. As $\omega$ decreases, the contributions of the smaller node and edge centralities to the edge and node centralitites respectively in Eqs 12 and 13 increase until the linear case when $\omega = 1$ and all centralities contribute their exact value. Further, as $\omega \to 0$ the node and edge centralities tend to unit contributions to the edge and node centralitites respectively equalling the ranking given by $P : (0, b, 0, d)$. This motivates the approach of defining a parameter space for $P : (a, b, c, d)$, for example $a = c = \omega$ and $b = d = 1/\omega$ where $\omega \in (0, 95]$ for $P : (\omega, 1/\omega, \omega, 1/\omega)$, and performing a search over this space to find parameters $a, b, c, d$ which permit a better performing model for a particular set of data.

In this section we have considered the three functions sets proposed in [12] and introduced the generalised power function set. Table 1 provides a summary of function sets discussed including the centrality calculations and interpretation.

## NSVC application to S.Cerevisiae

**S. Cerevisiae dataset.** The S.Cerevisiae protein complex dataset was obtained from [27]. The data contained in the authors dataset, *Complex_745*, combines four existing protein complex sets (CM270 [32], CM425 [32–34], CYC408 and CTC428 both from CYC2008 [35, 36]). Proteins that do not belong to a protein complex are not included in this analysis. Protein essentiality data was obtained from [25] and originated from MIPS [32], SGD [34], DEG [37] and SGDP [38]. Three single member complexes were removed from the original dataset with the resulting dataset containing 742 unique complexes of cardinality 2 or greater and 2166 proteins. The proteins are represented by the hypergraph nodes and the protein complexes by the hypergraph edges. To align with the existence and uniqueness conditions detailed in Theorem 1, we restrict our analysis to the largest of 113 connected components. This component

**Table 1. Summary of *f*, *g*, *φ*, *ψ* function sets with centrality calculations and description for unweighted hypergraphs.**

| Function Set | Functions | | | | Node centrality | | Edge centrality | |
|---|---|---|---|---|---|---|---|---|
| | $f$ | $g$ | $\varphi$ | $\psi$ | $x_i = \frac{1}{\lambda}\left(\sum\limits_{\substack{e \in E \\ i \in e}} y_e^a\right)^b$ | Type | $y_e = \frac{1}{\mu}\left(\sum\limits_{i \in e} x_i^c\right)^d$ | Type |
| $P : (a, b, c, d)$ | $x^a$ | $x^b$ | $x^c$ | $x^d$ | - | | - | |
| $P : (1, 1, 1, 1)$ *Linear T&H* | $x$ | $x$ | $x$ | $x$ | $\frac{1}{\lambda}\sum\limits_{\substack{e \in E \\ i \in e}} y_e$ | Lin | $\frac{1}{\mu}\sum\limits_{i \in e} x_i$ | Lin |
| $P : (0, 1, 0, 1)$ | 1 | $x$ | 1 | $x$ | $\frac{1}{\lambda}\sum\limits_{\substack{e \in E \\ i \in e}} 1$ | Deg | $\frac{1}{\mu}\sum\limits_{i \in e} 1$ | Deg |
| $P : (0, b, 0, d)$ | 1 | $x^b$ | 1 | $x^d$ | $\frac{1}{\lambda}\left(\sum\limits_{\substack{e \in E \\ i \in e}} 1\right)^b$ | Deg-power | $\frac{1}{\mu}\left(\sum\limits_{i \in e} 1\right)^d$ | Deg-power |
| $P : (95, 1/95, 95, 1/95)$ | $x^{95}$ | $x^{1/95}$ | $x^{95}$ | $x^{\frac{1}{95}}$ | $\frac{1}{\lambda}\left(\sum\limits_{\substack{e \in E \\ i \in e}} y_e^{95}\right)^{\frac{1}{95}}$ | Max | $\frac{1}{\mu}\left(\sum\limits_{i \in e} x_i^{95}\right)^{\frac{1}{95}}$ | Max |
| $P : (1, 1, 95, 1/95)$ *Max T&H* | $x$ | $x$ | $x^{95}$ | $x^{\frac{1}{95}}$ | $\frac{1}{\lambda}\sum\limits_{\substack{e \in E \\ i \in e}} y_e$ | Lin | $\frac{1}{\mu}\left(\sum\limits_{i \in e} x_i^{95}\right)^{\frac{1}{95}}$ | Max |
| *Log-exp T&H* | $x$ | $x^{1/p+1}$ | $ln(x)$ | $e^x$ | $\frac{1}{\lambda}\left(\sum\limits_{\substack{e \in E \\ i \in e}} y_e\right)^{\frac{1}{p+1}}$ | Lin-root | $\frac{1}{\mu}\prod\limits_{i \in e} x_i$ | Prod |

T&H: Function sets proposed in [12].

Lin: node/edge centrality is proportional to the sum of the centralities of incident edges/nodes respectively.

Deg: node/edge centrality is proportional to the node/edge degree respectively.

Max: node/edge max centrality is proportional to the maximum centrality of the incident edges/nodes respectively.

Deg-power: node/edge centrality is proportional to a power of the node/edge degree respectively.

Lin-root: node/edge centrality is proportional to a root of the sum of the centralities of incident edges/nodes respectively.

Prod: node/edge centrality is proportional to the product of the centralities of the incident edges/nodes respectively.

For $P : (a, b, c, d)$, $a, c \in \mathbb{R}_{\geq 0}$ and $b, d \in \mathbb{R}_{>0}$. For log-exp, $p \in \mathbb{Z}_{>0}$. For all node/edge centrality calculations, $\lambda, \mu \neq 0$.

contains 1739 proteins, 80% of the original proteins, and 558 protein complexes, 75% of the original complexes.

Unlike protein essentiality, there is no one defined measure of "complex essentiality". One approach has been to consider a (human) protein complex as "potentially" essential if it contains at least one essential protein [3]. Alternatively, in [39], the analysis of the core proteins gives rise to another definition, that a specific complex is essential if at least 60% of the core proteins are essential. Influenced by the latter definition we consider complexes as a whole which contain at least 60% essential nodes as essential but recognise that this is a simplification. In subsequent works, consideration of the complex core would likely yield interesting results.

The dataset contained 46 proteins with unknown essentiality, these were excluded reducing the set to 1693 proteins and 558 complexes. The final set contains 693 essential proteins and 230 essential complexes. For simplicity, we refer to the nodes representing essential proteins as essential nodes and equivalently essential edges are those representing essential complexes.

**Evaluation methods.** The S.Cerevisiae PCH incidence matrix *B* along with specified *f*, *g*, *φ*, *ψ* are used to calculate centralitites through Eq 1. There are no defined weights thus *N* and *W* are both identity matrices. The nodes and edges are ranked in descending order of their resulting centralities.

To evaluate the performance of the different function sets on essential protein identification, the *t* top ranked nodes are first identified. The value of *t* is based on a percentage of the

total number of essential proteins in the dataset, namely 5%, 10%, 25%, 50%, 75% and 100%. The ranking is then analysed against the hypothesis that a node $i$ with rank $r(i)$ is essential if $r(i) \leq t$. Given a dataset with $e_p$ essential proteins, a perfect classification model would have $r(i) \leq e_p$ for all essential nodes and a nonessential node would have $r(i) > e_p$. Essential nodes correctly classified as essential are considered *true positives* (TP). *False positives* (FP) are the nonessential nodes with $r(i) \leq t$ thus classified as essential. Clearly, $t = TP + FP$. Essential nodes with $r(i) > t$ are *false negatives* (FN) and nonessential nodes with $r(i) > t$ are *true negatives* (TN). An analogous approach is taken for analysing the performance of the chosen function sets when classifying essential edges, with $e_c$ being the total number of essential complexes in a dataset.

*Recall* denotes the ratio between the number of correctly classified essential proteins (equivalently, complexes) and the total number of essential proteins (equivalently, complexes) in the whole dataset: $\frac{TP}{TP+FN}$. *Precision* denotes the ratio between the number of correctly classified essential proteins (equivalently, complexes) and the number selected as essential: $\frac{TP}{TP+FP}$. Clearly, for an ideal classifier the precision would be 1.0 for $t \leq e_p$ (equivalently, $t \leq e_c$ for complexes), after which it declines until it reaches the percentage of essential proteins (equivalently, complexes) in the whole set. Mapping precision against recall for all values of $1 \leq t \leq 1693$ results in the precision recall curve.

The area under the precision-recall curve (PR-AUC) gives a single measure to compare classifiers. The better a classifier performs, the higher the PR-AUC will be. A perfect classifier will have a PR-AUC of 1.

We also consider the cumulative frequency of true positives for $1 \leq t \leq 1693$.

In both the precision recall curve and cumulative frequency graphs the ratio of essential to total number of proteins/complexes is used to define a baseline representing a random classifier to compare the performance of the chosen function sets.

These methods are similar to those used in previous papers such as [27] when identifying essential proteins in PINs.

## Results

First, we analyse the performance of the model with selected function sets for six thresholds and then consider its overall performance. Next, we analyse the associated classifications of edges. Finally, we consider a comparison with the LBBC graph centrality measure applied to a PIN.

### Performance of the NSVC model for essential protein classification in S. Cervisiae

**Classification performance for six thresholds.** The six thresholds used are $t \in \{34, 69, 173, 346, 519, 693\}$. These values equal 5%, 10%, 25%, 50%, 75% and 100% of $e_p = 693$, the number of essential proteins in the dataset.

*Set A: Base functions.* To investigate the centrality-lethality rule we apply the node and edge degree centralities using $P : (0, 1, 0, 1)$ to the S.Cervisiae PCH. Tudisco and Higham's max function set, redefined as $P : (1, 1, 95, 1/95)$, the linear function set, redefined as $P : (1, 1, 1, 1)$ and the log-exp function are also applied to investigate if essential nodes match the criteria required to be ranked higher using these functions. For ease we use $p = 14$ in the log-exp function set, giving $g(x) = x^{1/15}$, as this ensures a less skewed distribution of the centrality sizes towards the extremes of the interval [0, 1] than for smaller values of $p$. The function set $P : (1, 1, 46, 29/46)$ is also included, the motivation for the inclusion of this set is discussed within the following results.

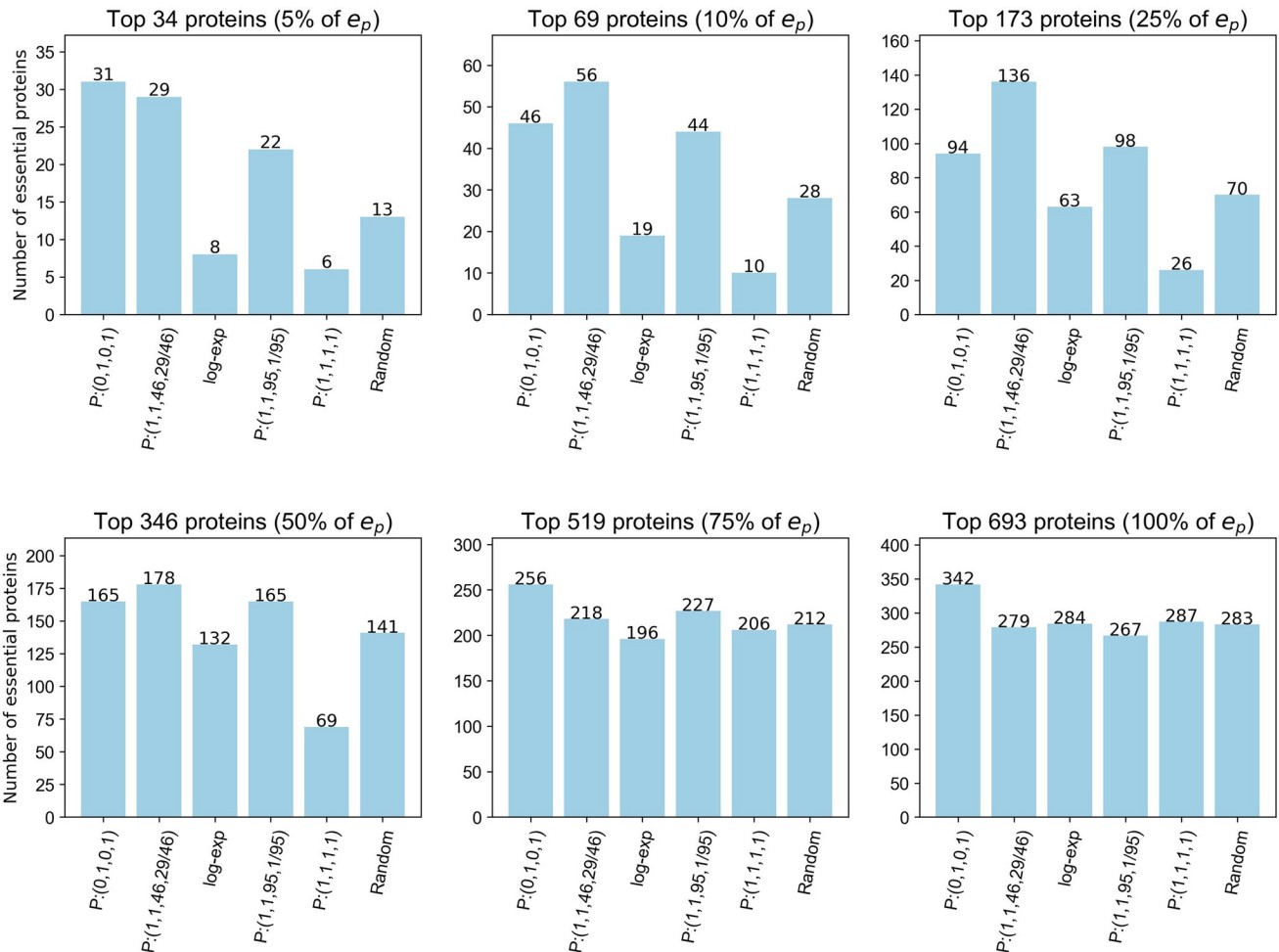

**Fig 1. Number of essential proteins correctly classified by the NSVC model using set A function sets.** Thresholds are based on a percentage of $e_p$, the total number of essential proteins in the PCH.

Fig 1 shows that using the function set $P : (0, 1, 0, 1)$ predicts more essential proteins than $P : (1, 1, 1, 1)$ and the log-exp function sets for the nominated thresholds. However the function set $P : (1, 1, 95, 1/95)$ performs similarly to $P : (0, 1, 0, 1)$ for the top 69, 173 and 346 proteins thus we consider building on this function set to surpass the performance of $P : (0, 1, 0, 1)$. As detailed in the Max function set section, $P : (1, 1, 95, 1/95)$ provides a node centrality that is proportional to a linear combination of the centralities of the edges to which it belongs and an edge centrality based only on the maximum centrality of its incident nodes. Utilising Eq 10 we see that $P : (1, 1, 95, 1/95)$ gives

$$\mu\lambda x_i = \sum_{\substack{e \in E \\ i \in e}} \Big(\sum_{\substack{j \in V \\ j \in e}} x_j^{95}\Big)^{1/95}.$$

Since $\omega = 95$ is large enough this gives:

$$\mu\lambda x_i \approx \sum_{\substack{e \in E \\ i \in e}} max\{x_{j1}, \ldots, x_{jk}\} \text{ where } e = \{x_{j1}, \ldots, x_{jk}\}.$$

**Table 2. A subset of functions included in set B.**

| Function Set | Node centrality type | Edge centrality type |
|---|---|---|
| P:(95,1/95,95,1/95) | max | max |
| P:(95,1/95,0,1) | max | degree |
| P:(95,1/95,1,1) | max | linear |
| P:(0,1,95,1/95) | degree | max |
| P:(0,1,1,1) | degree | linear |
| P:(1,1,0,1) | linear | degree |

However, if a node's centrality is not based on the maximum ranked node in each adjacent edge but on the centralities of a set of the top ranking nodes, then this motivates the use of function sets of the form $P : (1, 1, c, k/c)$. To investigate this we considered $P : (1, 1, c, k/c)$ where $c, k \in \{0.1, 0.2, \ldots 0.9\} \cup \{1, 2, \ldots 95\}$. The top performing function sets for each of the six thresholds were identified and those which were also top performers for predicting essential complexes were singled out. From this group, the function set $P : (1, 1, 46, 29/46)$ was highlighted as it had the highest node PR-AUC. This set exceeds the number of essential proteins predicted by $P : (0, 1, 0, 1)$ and $P: (1, 1, 95, 1/95)$ for $t \in \{69, 173, 346\}$ (Fig 1).

*Set B: Combined functions.* A second approach taken was to combine and extend the degree, linear and max functions through an exhaustive search of the combinations for $P : (a, b, c, d)$ with $a, c \in \{0, 1, 1/95, 95\}$ and $b, d \in \{1, 1/95, 95\}$ to see if any of these combinations performed better than the set A functions. Included in these functions are the intuitive approaches summarised in Table 2. These functions do not necessarily align with underlying characteristics of the PCH dictating essentiality but they are being showcased here as important considerations for other applications. For example, $P : (0, 1, 95, 1/95)$ ranks nodes by node degree centrality, utilising the number of edges a node belongs to, whilst simultaneously ranking edges by max centrality, utilising the maximum centrality of the nodes in the edge. Table 1 provides further detail of these node and edge centralities.

For function sets of this form, $P : (1/95, 95, 1, 1/95)$ was identified as being a top performer using the same approach as that which identified $P : (1, 1, 46, 29/46)$ in set A. Fig 2 shows the results for the function sets summarised in Table 2 together with $P : (1/95, 95, 1, 1/95)$. As expected from the discussion in the Generalised power function set section, $P : (0, 1, 1, 1)$ and $P : (0, 1, 95, 1/95)$ produce the same node rankings as $P : (0, 1, 0, 1)$ as they are of the form $P : (0, b, c, d)$.

**Overall classification performance.** We now compare the overall performance of the top performers described in the previous section alongside $P : (1, 1, 1, 1)$ and $P : (1, 1, 95, 1/95)$.

Fig 3 shows clearly that $P : (0, 1, 0, 1)$ and $P : (1, 1, 95, 1/95)$ perform approximately equal at first but then $P : (1, 1, 95, 1/95)$ drops off significantly and predicts less than or equal to random for larger values of $t$. This later poorer performance for $P : (1, 1, 95, 1/95)$ results in a PR-AUC of 0.45, only marginally above the random PR-AUC of 0.40. The PR-AUC for $P : (0, 1, 0, 1)$ is higher at 0.51. Function set $P : (1/95, 95, 1, 1/95)$ performs approximately equal to $P : (0, 1, 0, 1)$ reflected in an identical PR-AUC of 0.51. The performance of $P : (1, 1, 46, 29/46)$, measured by a PR-AUC of 0.52, is only marginally better than $P : (0, 1, 0, 1)$ however its performance for smaller values of $t$ is significantly better (Figs 3 and 4). It is this that sets it aside from the others. Being able to identify small sets of predominantly essential proteins could be useful in many small scale experiments.

The $P : (1, 1, 1, 1)$ function set has the lowest PR-AUC of 0.38 with predictive capabilities less than random in the upper thresholds. The overall poor performance of this function set

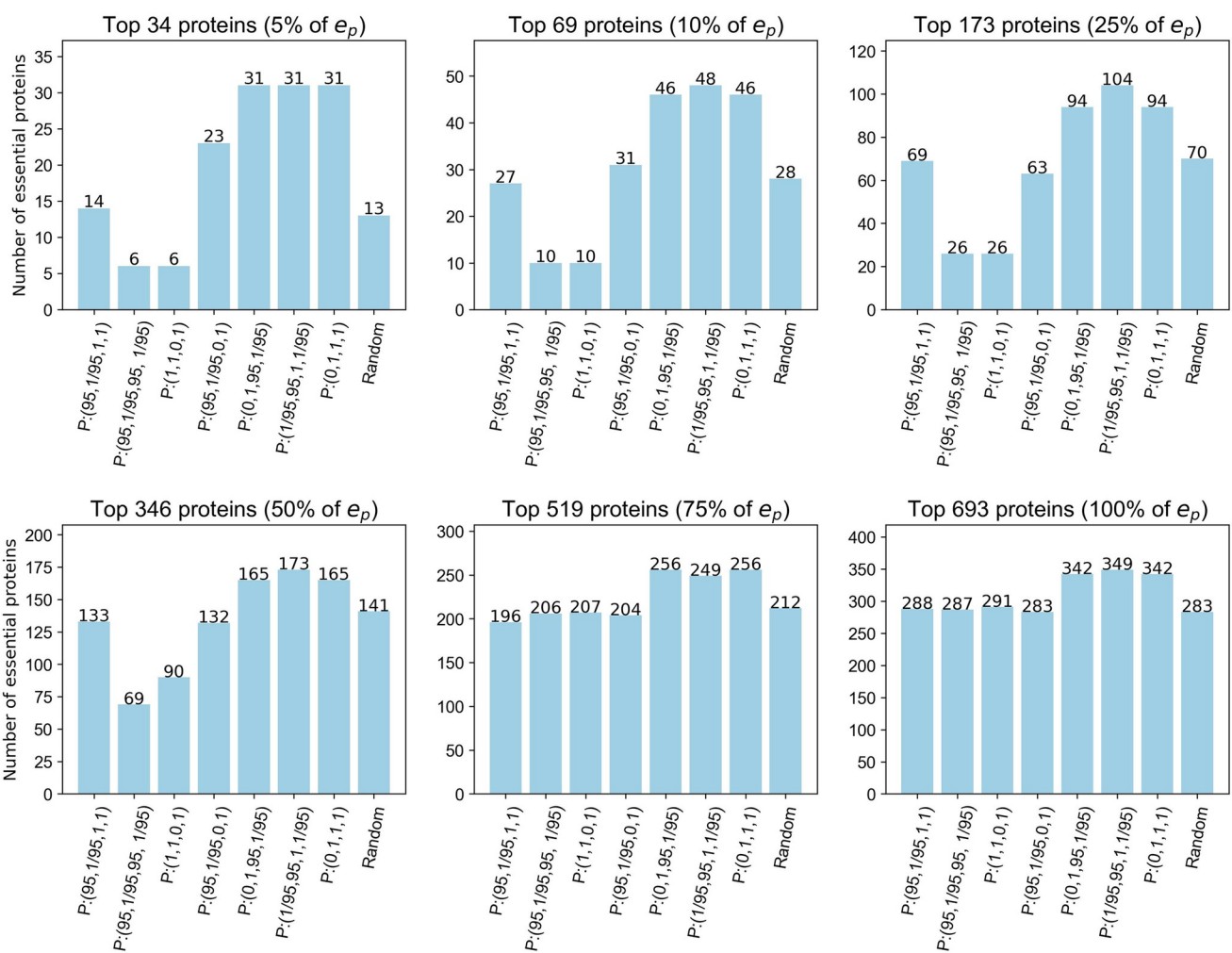

**Fig 2. Number of essential proteins correctly classified by the NSVC model using a subset of set B function sets.** Thresholds are based on a percentage of $e_p$, the total number of essential proteins in the PCH.

indicates that the network characteristics of the essential proteins in the PCH do not align well with those influencing a higher ranking under $P : (1, 1, 1, 1)$. Specifically, the essentiality of a protein cannot be accurately predicted solely by the number and centrality of the complexes it belongs to (Eq 2) or equivalently, by the number and centrality of the proteins in those complexes (Eq 10).

## Performance of the NSVC model for essential edge classification

The focus so far has been primarily on node classification however a significant advantage of the NSVC model over previous models is that it simultaneously provides edge rankings. In this section we briefly look at the performance of select functions identified for node classification and interpret their performance when they are applied to classify edges.

The six thresholds nominated are $t \in \{11, 23, 57, 115, 172, 230\}$ corresponding to 5%, 10%, 25%, 50%, 75% and 100% of $e_c = 230$, the number of essential complexes in the dataset. We see in Fig 5 that $P : (1, 1, 46, 26/46)$ again performs best for smaller values of $t$.

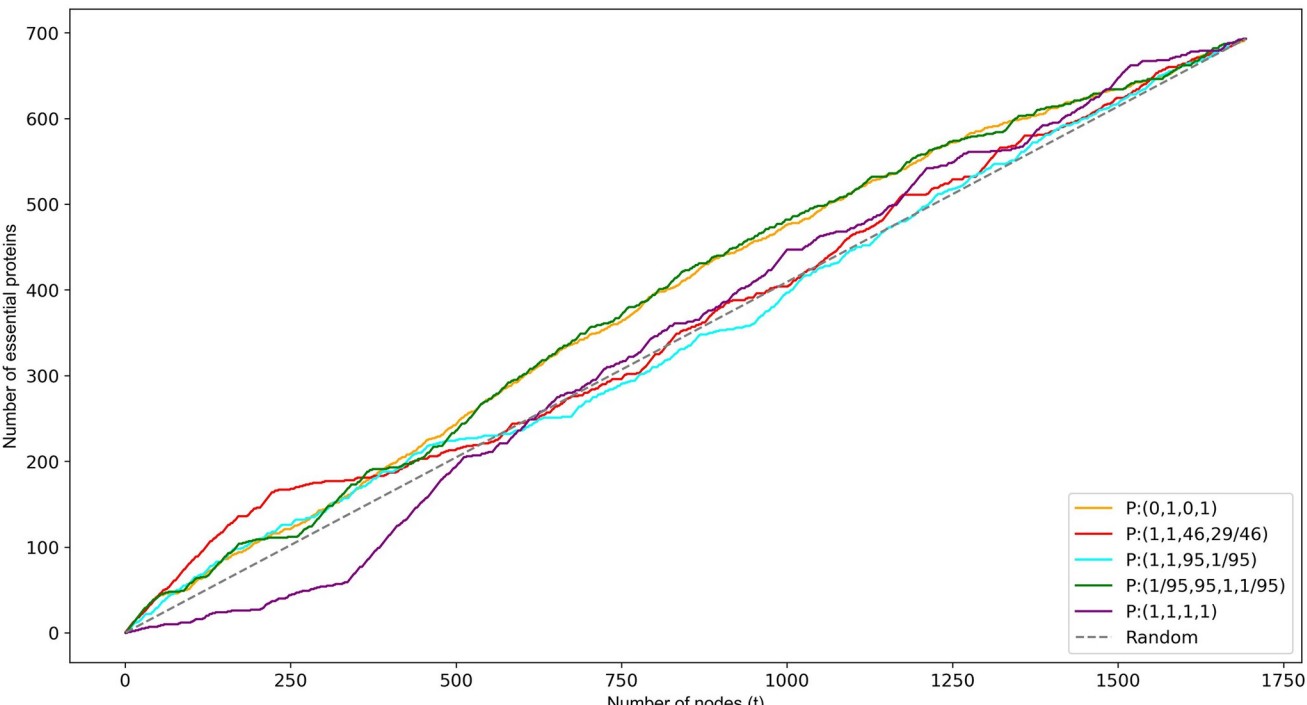

**Fig 3. Cumulative frequency of the number of essential proteins correctly classified via the top _t_ ranked nodes.**

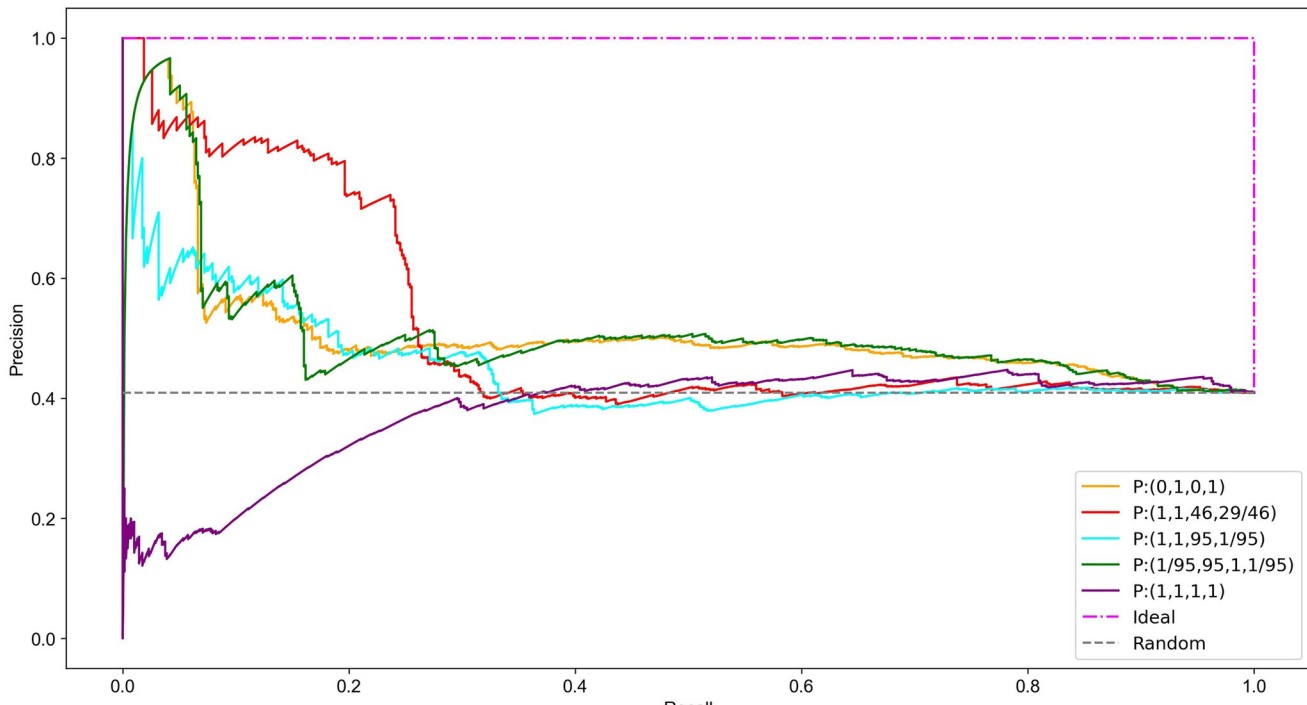

**Fig 4. Comparison of the precision versus recall for essential protein classification.**

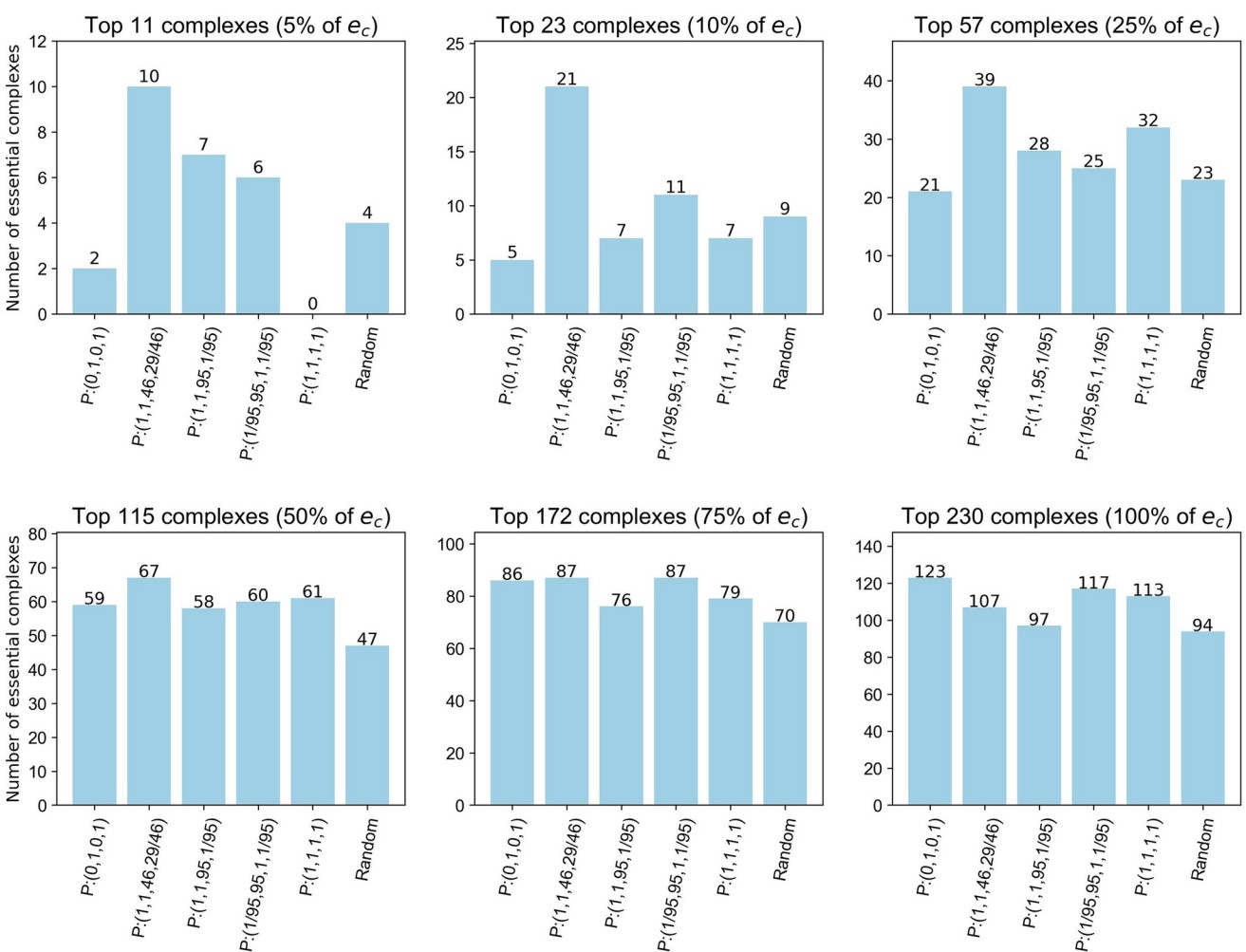

**Fig 5. Number of essential complexes correctly classified by the NSVC model using nominated function sets.** Thresholds are based on a percentage of $e_c$, the total number of essential complexes in the PCH.

We can see in Fig 6 that the function set $P : (0, 1, 0, 1)$ is the best performer in the middle ranges and $P : (1/95, 95, 1, 1/95)$ performs best for larger values of $t$. Overall, the highest PR-AUC is 0.53 resulting from $P : (1, 1, 46, 29/46)$ followed by 0.47 for $P : (0, 1, 0, 1)$ compared to a random PR-AUC of 0.41. The precision recall curve, Fig 7, depicts the varying PR-AUC results with $P : (1, 1, 46, 29/46)$ clearly dominating.

Although it is likely that other variations of $P : (a, b, c, d)$ would yield additional positive results it is motivating that $P : (1, 1, 46, 29/46)$ predicts both a higher number of essential proteins and complexes than the other trialed function sets including the degree based $P : (0, 1, 0, 1)$ for smaller values of $t$.

## Comparison with a classification method applied to a PIN

Although the additional ranking of complexes gives the NSVC a clear advantage over PIN-based methods, it is still worth comparing its performance when ranking essential proteins to a comparable PIN-based measure.

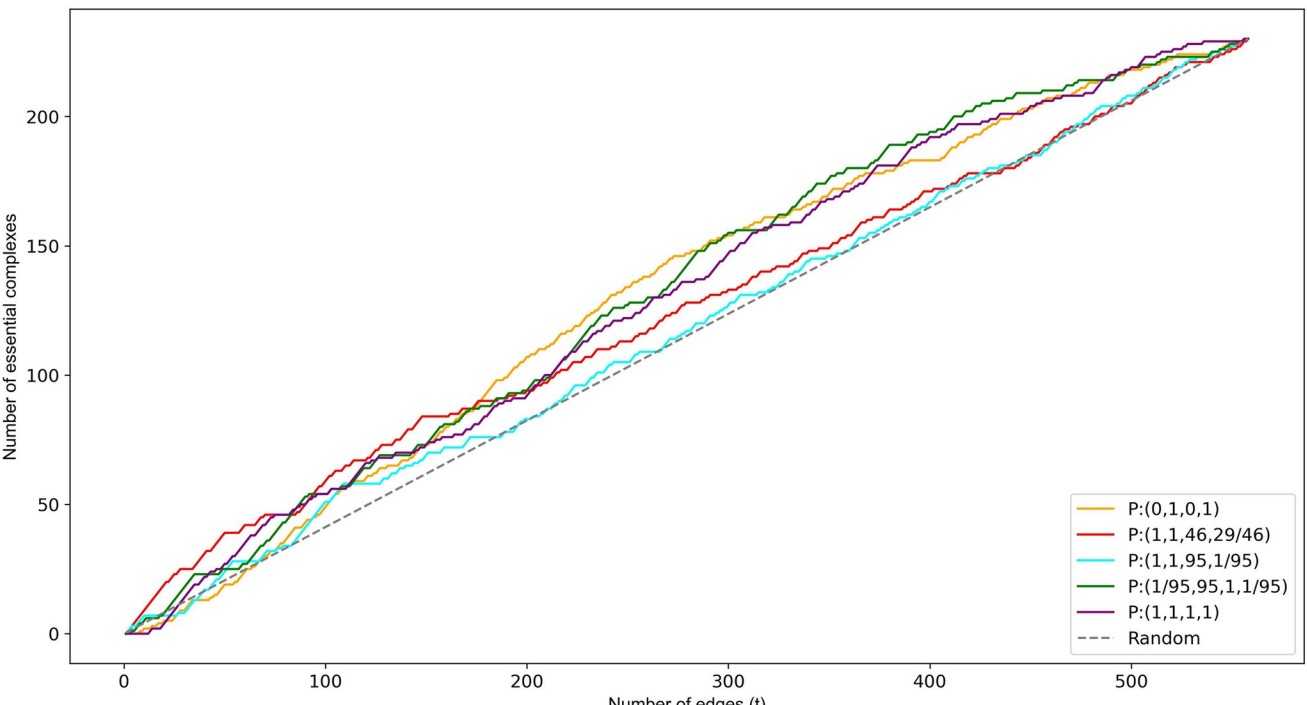

**Fig 6. Cumulative frequency of the number of essential complexes correctly classified via the top *t* ranked edges.**

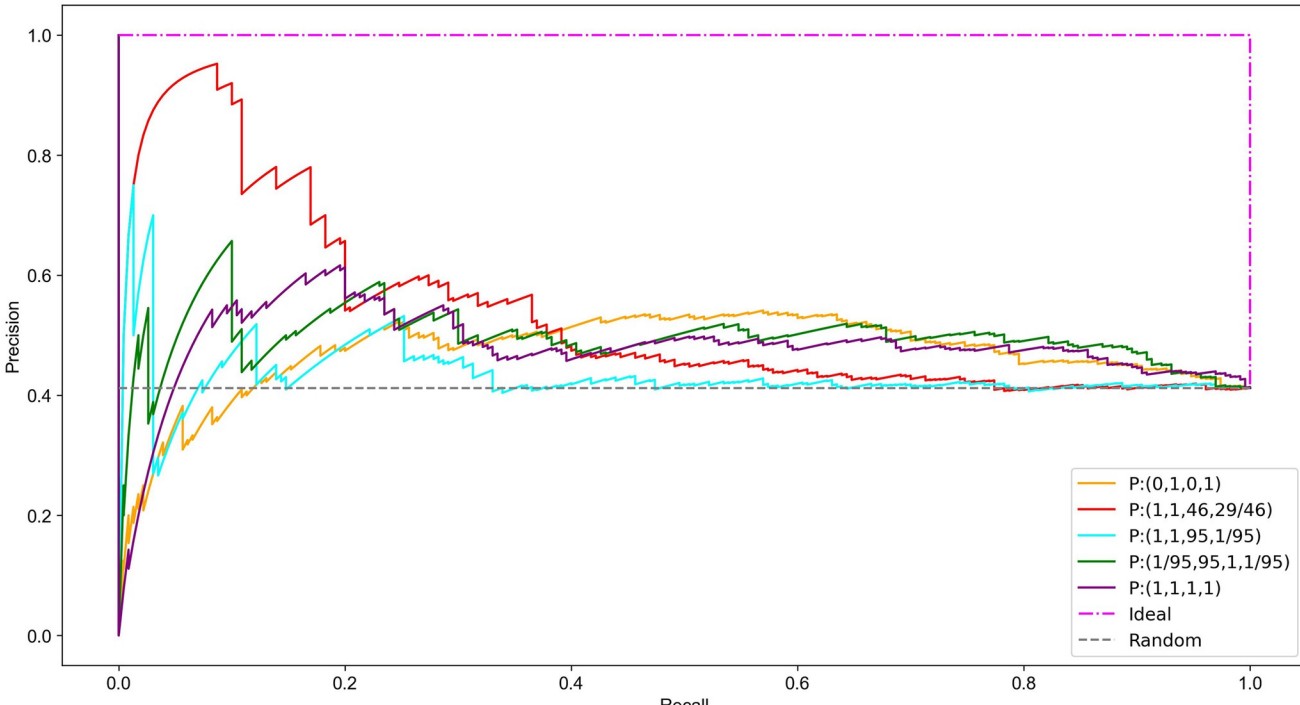

**Fig 7. Comparison of the precision versus recall for essential protein complex classification.**

The performance of a classification method applied to a PIN varies depending on the PIN chosen and the threshold being considered. Focusing only on those which included protein interaction and protein complex data in their calculations to align with the NSVC model, we compared the methods applied in [25–27]. Of these, LBBC [25] correctly classified the largest number of essential proteins in almost all thresholds across all PINs utilised in these papers. The best performance was obtained when applied to the MIPS PIN [32]. This PIN contains 4246 proteins of which 1016 are essential, 3195 are nonessential and 335 are unknown. The LBBC centralities used in the following comparison were obtained from code provided in [25].

Applying both the LBBC and NSVC methods to the same dataset for comparison is not feasible as the LBBC method uses the structure of the PIN in its calculations whilst the NSVC model uses the different structure of the PCH in its calculations. The number of proteins in the MIPS PIN and the PCH differ with neither being a subset of the other and, significantly, the prevalence of essential proteins differ. The MIPS PIN contains 24% essential proteins compared to 41% in the PCH which introduces complications when comparing the classifiers.

However, we can consider the intersection between the MIPS proteins and the PCH proteins and rank these proteins based on their original order from the LBBC and NSVC applications. The intersection contains 1409 nodes, of which 589 are essential and 820 are nonessential. For the NSVC model the three top performing function sets: $P : (0, 1, 0, 1)$, $P : (1/95, 95, 1, 1/95)$ and $P : (1, 1, 46, 29/46)$ are used. The resulting rankings are analysed using the same methods applied above.

Fig 8 shows that $P : (1, 1, 46, 29/46)$ continues to perform strongest for the 5%, 10% and 25% thresholds with LBBC exceeding its performance for the 50%, 75% and 100% thresholds.

## Discussion and conclusion

Research identifying essential proteins through their interactions and associated centralities has been predominantly focused on using graph measures applied to PINs. Our research has shown there is merit in using a PCH and applying newer hypergraph centrality measures, specifically the NSVC model. This model has advantages over many existing node centrality measures applied to PINs. The calculations are more direct and adaptable and, significantly, it provides for both the ranking of proteins and complexes.

In our novel application of this model we identified function sets which performed better than random for both the classification of essential proteins and essential protein complexes. We highlighted the above average performance of the node-edge degree centrality, $P : (0, 1, 0, 1)$, evidencing the centrality-lethality rule in a hypergraph setting and extended the NSVC model to a generalised function set $P : (a, b, c, d)$. This flexible approach resulted in the identification of $P : (1, 1, 46, 29/46)$ which gave the best classification performance for a range of thresholds and competed with a leading PIN-based measure, LBBC [25].

In this work we have mainly focused on the node centralities and the associated classification of essential proteins. The next step would be to broaden the results by refocusing on edge centralities and protein complexes. This could include alternative definitions of complex essentiality such as essentiality based on the core of a protein complex as discussed in [39]. A growing body of research indicates that the essentiality of a protein is linked to the essentiality of the complex to which it belongs as opposed to being a characteristic solely of the protein itself [39–41]. This further motivates shifting the focus onto edges. Research also shows that essential proteins are more likely to belong to complexes [40]. This is evident in the PCH dataset used in this research which is denser in essential proteins than the MIPS dataset for example. As protein complex data becomes more readily available it is anticipated that methods like the NSVC model will prove valuable in this research area.

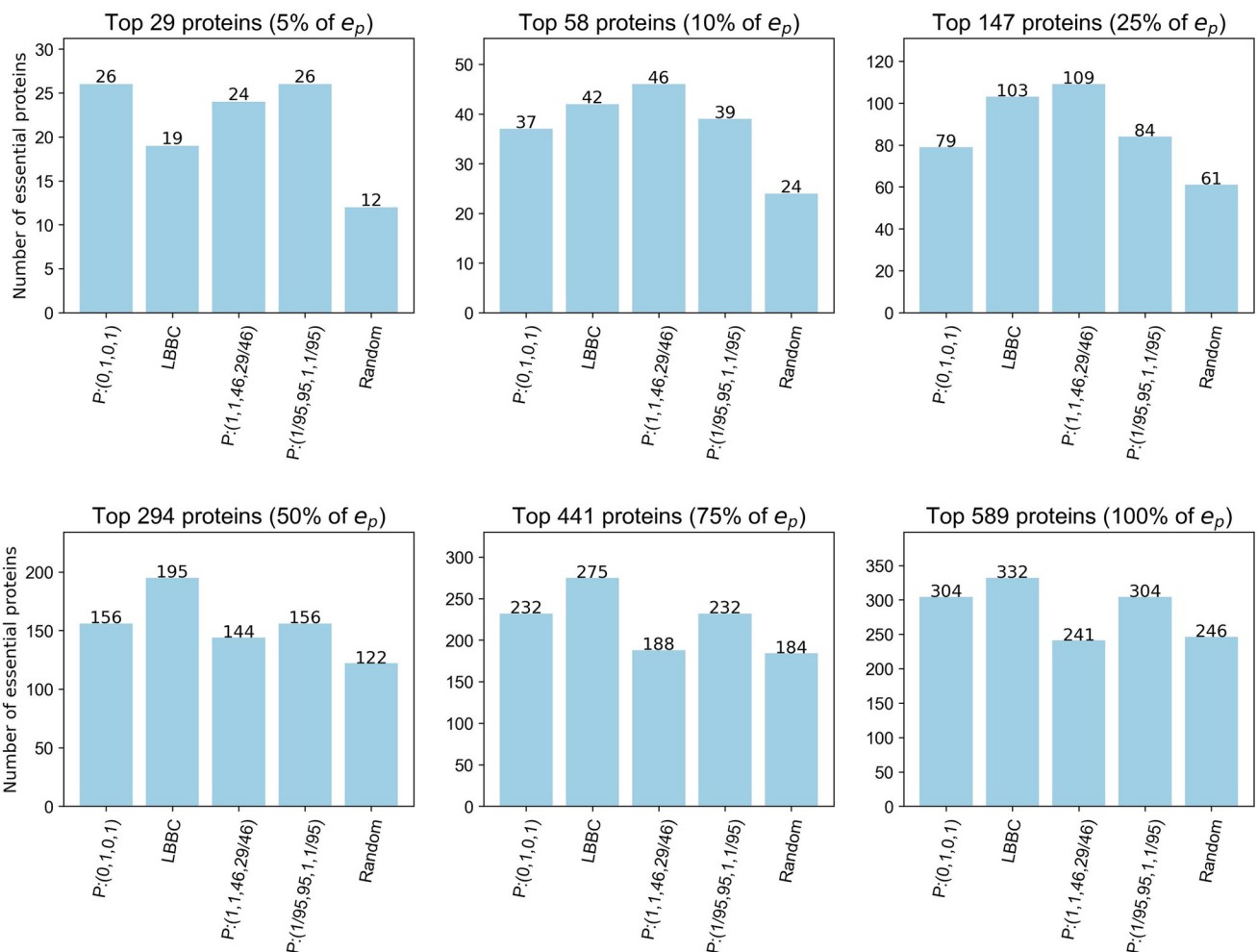

**Fig 8. Number of essential proteins correctly classified by the LBBC method and the NSVC model applied to the PIN-PCH intersection dataset.**
Thresholds are based on a percentage of $e_p$, the total number of essential proteins in the PIN-PCH intersection.

Recently Tudisco and Higham proposed a core-periphery score vector that assigns large values to nodes in the core of a hypergraph and small values to those in the periphery [42]. Although out of scope of this work and requiring input from proteomic experts, this measure would be interesting to analyse due to proposed connections between hypergraph cores and essential proteins and complexes such as those discussed in [30]. Other centralities which provide insight into a nodes interaction with the edges to which it belongs could also be considered for application to a PCH. In particular, subgraph centrality [43], which assigns a node ranking based on closed walks originating from the node, and vector centrality [44], which gives a vector communicating a nodes interaction with edges of different cardinality.

Factors influencing the 'importance' of a node or edge in a biological hypernetwork, such as the PCH studied in this work, aren't necessarily topological. With input from experts in the biological field, non-topological information could be incorporated into the NSVC model. For instance, weight functions could be utilised to represent quantifiable node and edge data or directed hypergraphs could be used to model interactions more accurately.

In this paper we have showcased an alternative approach to identifying essential proteins by proposing the use of the NSVC model applied to a PCH. Our approach also has the added

advantage of simultaneously providing protein complex rankings. The potential of extending this application to other biological hypernetworks is self-evident. It is hoped that the work initiated in this paper motivates further research into using hypergraph centrality, particularly the NSVC model, to analyse complex biological processes.

## Author Contributions

**Conceptualization:** Sarah Lawson, Diane Donovan, James Lefevre.

**Data curation:** Sarah Lawson.

**Formal analysis:** Sarah Lawson, Diane Donovan, James Lefevre.

**Funding acquisition:** Diane Donovan.

**Investigation:** Sarah Lawson, Diane Donovan, James Lefevre.

**Methodology:** Sarah Lawson, Diane Donovan, James Lefevre.

**Project administration:** Sarah Lawson, Diane Donovan, James Lefevre.

**Resources:** James Lefevre.

**Software:** Sarah Lawson.

**Supervision:** Diane Donovan, James Lefevre.

**Validation:** Sarah Lawson, Diane Donovan, James Lefevre.

**Visualization:** Sarah Lawson, Diane Donovan, James Lefevre.

**Writing – original draft:** Sarah Lawson.

**Writing – review & editing:** Sarah Lawson, Diane Donovan, James Lefevre.

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
