## [Decision Letter · Decision Letter 0]

18 Jul 2024

PONE-D-24-22937An application of node and edge nonlinear hypergraph centrality to a protein complex hypernetworkPLOS ONE

Dear Dr. Lawson,

Thank you for submitting your manuscript to PLOS ONE. After careful consideration, we feel that it has merit but does not fully meet PLOS ONE’s publication criteria as it currently stands. Therefore, we invite you to submit a revised version of the manuscript that addresses the points raised during the review process. Both reviewers find your work to be very interesting and worthy of publication. However, there are several comments and suggestions that deserve your consideration, particularly those related to improving the clarity of the manuscript and the comparison, if possible, of your method with previous pairwise network approaches.  

We look forward to receiving your revised manuscript.

Kind regards,

Irene Sendiña-Nadal

Academic Editor

PLOS ONE

[ This work was supported by The Australian Research Council,through the Centre of Excellence for Plant Success

in Nature and Agriculture (CE200100015).]

 [This work was supported by The Australian Research Council, through the Centre of Excellence for Plant Success

in Nature and Agriculture (CE200100015) https://www.plantsuccess.org/. Author DD is CI in the CoE. The CoE and ARC had no direct role in study design, data collection and analysis, decision to publish, or preparation of the manuscript.]

4. We noted in your submission details that a portion of your manuscript may have been presented or published elsewhere. [Paper on arXiv: https://arxiv.org/abs/2406.01880 preprint server only] Please clarify whether this conference proceeding was peer-reviewed and formally published. If this work was previously peer-reviewed and published, in the cover letter please provide the reason that this work does not constitute dual publication and should be included in the current manuscript.

Reviewers' comments:

Reviewer's Responses to Questions

**Comments to the Author**

1. Is the manuscript technically sound, and do the data support the conclusions?

Reviewer #1: Yes

Reviewer #2: Yes

2. Has the statistical analysis been performed appropriately and rigorously? 

Reviewer #1: Yes

Reviewer #2: Yes

3. Have the authors made all data underlying the findings in their manuscript fully available?

Reviewer #1: Yes

Reviewer #2: Yes

4. Is the manuscript presented in an intelligible fashion and written in standard English?

Reviewer #1: Yes

Reviewer #2: Yes

5. Review Comments to the Author

Reviewer #1: The paper deals with using centrality measures to find essential proteins and essential protein complexes when they interact. Traditionally proteins are interpreted as nodes of a network while interactions are understood as edges between two nodes in that network (PIN). The novelty in this paper is that interactions are not determined by just a pair of proteins but by a group of them, and this indeed justifies the use of hypergraphs. With applications in view the authors introduce some sound adaptations on the Tudisco-Higham NSVC model in order to obtain the ranking vector. The paper is clearly written and the model consistently used to achieve their goal on the S. Cervisiae dataset. Therefore I recommend publication.

Still there are some minor comments referring to the generalized power function set introduced by the authors.

1) The role of zero as a value for $a,b$ deserves some additional explanation, especially when a comparison with the three examples given by Tudisco-Higham is made.

2) The authors observe that the rankings obtained don't vary when the variables $a,b,c,d$ belong to the same family, but I miss some addtional justification of why this is so in the examples given.

3) A natural question arises about the vector structure in $\\mathbb{R}^4$ of those families above. Perhaps some comment or hint would be welcome in this direction but I leave it to the authors' choice.

Reviewer #2: This article provides an interesting application of hypergraphs and an eigenvector/eigenvalue based centrality method to identify essential proteins and essential protein complexes. They apply the method to a protein complex hypergraph of S. Cerevisiae. The application of the method and the evaluation were straightforward.

I found the article fairly easy to read. The details in section 2.2 were the most opaque mostly because there is a lot of text and a lot of notation. I'm a mathematician myself (not a biologist!) so this doesn't scare me. But I think it would make it easier to digest if there were a summary table of the different function configurations and power configurations and what they correspond to. You do have Table 1, which is helpful. But I think a more general table earlier on, in addition, would bring even more clarity. As you apply them in section 3 it becomes difficult to flip back and forth to recall intuition.

The reason that I put this as "major revision" rather than "minor" is the following: The authors point out in the first sentence of the discussion that "Research identifying essential proteins through their interactions has been predominantly focused on using graph measures applied to PINs." It would be nice to compare your findings to a comparable graph approach. Does the state of the art graph measure applied to PINs have even higher scores? This is important in convincing biology researchers that it's worth getting into all of this additional structure of hypergraphs and math of the NSVC measures. You could add it as a method to each of your figures as a baseline for comparison (like you have random as a baseline already). I understand the graph approach would only work for essential proteins and not essential complexes, which is already a point in your favor. But does NSVC do better at identifying essential proteins?

A couple of other minor comments:

- On page 2 when you define weighted node degree in terms of w(e) and weighted edge degree in terms of v(i) you haven't defined the w and v functions. I'd suggest to avoid v as a weight function since v_i are vertices.

- When you introduce the NSVC model please provide a bit more intuition at the front. How should we think of the f, g, phi, psi functions? What are x and y really capturing?

6. PLOS authors have the option to publish the peer review history of their article (what does this mean?). If published, this will include your full peer review and any attached files.

Reviewer #1: No

Reviewer #2: No

---

## [Author Response · Author response to Decision Letter 0]

23 Aug 2024

Response to Reviewers:

Reviewer #1:

1. The role of zero as a value for $a,b$ deserves some additional explanation, especially when a comparison with the three examples given by Tudisco-Higham is made.

o Further clarification has been added to manuscript lines 213-214 (revised manuscript with track changes lines 230-231) for b=d=0 and calculations and explanation included in Table 1 for P:(0,b,0,d) (i.e. a=c=0).

2. The authors observe that the rankings obtained don't vary when the variables $a,b,c,d$ belong to the same family, but I miss some additional justification of why this is so in the examples given.

o Clarification has been added to manuscript lines 228-230 (revised manuscript with track changes lines 248-250) with calculations and explanations included in Table 1.

3. A natural question arises about the vector structure in $\\mathbb{R}^4$ of those families above. Perhaps some comment or hint would be welcome in this direction but I leave it to the authors' choice.

o This is solely notation. Explanation has been rewritten in manuscript line 212 (revised manuscript with track changes line 229).

Reviewer #2:

1. I think it would make it easier to digest if there were a summary table of the different function configurations and power configurations and what they correspond to.

o Table 1 has been added. (Please note that the recommended latexdiff used to track changes causes an issue with this table formatting in the ‘revised manuscript with track changes’ document however the table is correctly formatted in the ‘manuscript’ document). 

2. It would be nice to compare your findings to a comparable graph approach. Does the state of the art graph measure applied to PINs have even higher scores? … does NSVC do better at identifying essential proteins?

o Discussion regarding the issues involved in identifying a state-of-the-art graph measure applied to a PIN dataset and comparing this with the NSVC model applied to a PCH dataset have been included in the “Comparison with a classification method applied to a PIN” section beginning on manuscript line 399 (revised manuscript with track changes line 456). We feel that these issues make a sound comparison infeasible. However, if the reviewer still feels that it is critical for publication we have provided a comparison using the MIPS PIN-PCH intersection beginning on manuscript line 419 (revised manuscript with track changes line 476).

o Should the reviewer agree that the issues with the comparison warrant it infeasible then we propose the comparison is omitted.

3. On page 2 when you define weighted node degree in terms of w(e) and weighted edge degree in terms of v(i) you haven't defined the w and v functions. I'd suggest to avoid v as a weight function since v_i are vertices.

o Functions defined on manuscript lines 139-141 (revised manuscript with track changes lines 147-150).

o v(i) replaced by n(i) throughout manuscript.

4. When you introduce the NSVC model please provide a bit more intuition at the front. How should we think of the f, g, phi, psi functions? What are x and y really capturing?

o Further discussion regarding this has been added to manuscript lines 59-66 (revised manuscript with track changes lines 64-72).

The highlighted changes in the ‘Revised Manuscript with Track Changes’ document include the changes detailed above, changes required to meet formatting guidelines and changes made to improve readability.

---

## [Decision Letter · Decision Letter 1]

13 Sep 2024

An application of node and edge nonlinear hypergraph centrality to a protein complex hypernetwork

PONE-D-24-22937R1

Dear Dr. Lawson,

We’re pleased to inform you that your manuscript has been judged scientifically suitable for publication and will be formally accepted for publication once it meets all outstanding technical requirements.

Kind regards,

Irene Sendiña-Nadal

Academic Editor

PLOS ONE

Additional Editor Comments (optional):

Reviewers' comments:

Reviewer's Responses to Questions

**Comments to the Author**

1. If the authors have adequately addressed your comments raised in a previous round of review and you feel that this manuscript is now acceptable for publication, you may indicate that here to bypass the “Comments to the Author” section, enter your conflict of interest statement in the “Confidential to Editor” section, and submit your "Accept" recommendation.

Reviewer #1: All comments have been addressed

2. Is the manuscript technically sound, and do the data support the conclusions?

Reviewer #1: Yes

3. Has the statistical analysis been performed appropriately and rigorously? 

Reviewer #1: N/A

4. Have the authors made all data underlying the findings in their manuscript fully available?

Reviewer #1: Yes

5. Is the manuscript presented in an intelligible fashion and written in standard English?

Reviewer #1: Yes

6. Review Comments to the Author

Reviewer #1: I thank the authors for properly addressing all comments. I am happy to recommend publication now.

7. PLOS authors have the option to publish the peer review history of their article (what does this mean?). If published, this will include your full peer review and any attached files.

Reviewer #1: No

---

## [Editor Report · Acceptance letter]

23 Sep 2024

PONE-D-24-22937R1 

PLOS ONE

Dear Dr. Lawson, 

I'm pleased to inform you that your manuscript has been deemed suitable for publication in PLOS ONE. Congratulations! Your manuscript is now being handed over to our production team.

Kind regards, 

on behalf of

Dr. Irene Sendiña-Nadal 

Academic Editor

PLOS ONE